# Validation of revised patient measures of safety: PMOS-30 and PMOS-10

Gemma Louch  [ORCID],[1] Caroline Reynolds,[1] Sally Moore,[1] Claire Marsh,[1] Jane Heyhoe,[1] Abigail Albutt,[1] Rebecca Lawton  [ORCID],[1,2] on behalf of the Yorkshire Quality and Safety Research Group

[1]Yorkshire Quality and Safety Research Group, Bradford Institute for Health Research, Bradford, UK
[2]School of Psychology, University of Leeds, Leeds, UK

**Correspondence to**
Gemma Louch;
Gemma.Louch@bthft.nhs.uk

## ABSTRACT

**Objectives** There is growing evidence that patients can provide feedback on the safety of their care. The 44-item Patient Measure of Safety (PMOS) was developed for this purpose. While valid and reliable, the length of this questionnaire makes it potentially challenging for routine use. Our study aimed to produce revised, shortened versions of PMOS (PMOS-30 and PMOS-10), which retained the psychometric properties of the longer version.

**Participants** To produce a shortened diagnostic measure, we analysed data from 2002 patients who completed PMOS-44, and examined the reliability of the revised measure (PMOS-30) in a sample of 751 patients. To produce a brief standalone measure, we again analysed data from 2002 patients who completed PMOS-44, and tested the reliability and validity of the brief standalone measure (PMOS-10) in a sample of 165 patients.

**Methods** The process of shortening the questionnaire involved a combination of secondary data analysis (eg, Standard Deviation and inter-item correlations) and a consensus group exercise to produce PMOS-30 and examine face validity. Analysis of PMOS-30 data examined reliability (eg, Cronbach's alpha). Further secondary data analysis (ie, corrected item-total correlations) produced PMOS-10, and primary data collection assessed its reliability and validity (eg, Cronbach's alpha, analysis of variance).

**Results** Fourteen items were removed to produce PMOS-30 and the percentage of negatively worded items was reduced from 57% to 33%. PMOS-30 demonstrated good internal reliability ($\alpha$=0.89). The 10 items with the highest corrected item-total correlations across both PMOS-44 and PMOS-30 composed PMOS-10. PMOS-10 had good internal reliability ($\alpha$=0.79), demonstrated convergent validity; however, discriminant validity was not established.

**Conclusions** Two revised, shortened versions of the original PMOS-44 (PMOS-30 and PMOS-10) were produced to capture patient feedback about safety in hospital. The measures demonstrated good reliability and validity, and preserved the psychometric properties of the original measure.

## BACKGROUND

Measurement has posed an ongoing challenge in patient safety improvement. The difficulties and tensions associated with developing and implementing measures of patient safety are well recognised, and there is a need

### Strengths and limitations of this study

► Reliability testing of PMOS-30 took place in a variety of wards across three hospital trusts.
► The involvement of patient representatives is a key strength of the PMOS-10 development phase.
► A limitation of our work is that our studies only included patient participants who could understand English.

for measures to be relevant to multiple stakeholders, have a scientific basis, as well as being feasible and usable.[1] Indeed, there have been calls for more time and money to be invested in the science of safety measurement and the implementation of systems to capture such measures in order to understand whether safety is improving.[2] Furthermore, there is recognition that safety cannot be captured in a single measure, and that patients and carers play an essential role in safety monitoring, but are often an underused resource.[3]

The collection of feedback on patient satisfaction and experiences of care is now commonplace,[4] and in the UK, multiple high profile reports have highlighted the importance of listening and responding to patient views about safety.[5–7] There is growing evidence that patients can provide feedback on the safety of their care,[8 9] and research to suggest that patients can offer a unique perspective on patient safety not captured by other established methods of incident detection.[10] To provide a proactive diagnostic tool which systematically invites the patient perspective on safety, researchers from the UK developed an intervention to provide a theory and evidence-based approach for the collection of hospital in-patient feedback about safety to support service improvement. The Patient Reporting and Action for a Safe Environment (PRASE) intervention[11 12] includes two measurement tools. The first is a 44-item theory based measure—the Patient Measure of Safety (PMOS),[13 14] informed

by the domains of the Yorkshire Contributory Factors Framework (YCFF).[15] The second is the Patient Incident Reporting Tool (PIRT)[12] which enables patients to report safety concerns and positive experiences.

The original measure included a minimum of two items per domain, across nine domains, known to contribute to safety incidents in hospitals: (1) communication and teamworking; (2) organisation and care planning; (3) access to resources; (4) ward type and layout; (5) information flow; (6) staff roles and responsibilities; (7) staff training; (8) equipment (design and functioning) and (9) delays. Within a randomised controlled trial (RCT) of the PRASE intervention,[11] PMOS data were collected from over 2000 patient participants. The researchers and research nurses who facilitated this data collection reported that some patients, particularly those who were older or very unwell, struggled with the length of the measure and the phrasing of some of the negatively worded items. Evidence suggests that shorter measures and a reduction in assessment time can lessen the burden on participants and improve response rates and missing data.[16] Therefore, for a large number of questionnaires used in a healthcare context, often short form or brief measures are produced to address this issue with the aim of maintaining acceptable reliability and validity.[17 18]

While in the PRASE RCT[11] researchers were on hand to answer questions and encourage responses, so the above issues did not pose a major problem (response rates were 86%), these issues may result in lower uptake as use becomes more widespread. For example, if data are collected via hospital volunteers (one option that has been tested), difficulties with question wording and/or the length of the measure pose a threat to the reliability and validity of the responses, and may result in missing data. Indeed, in a formative evaluation of the implementation of PRASE with hospital volunteers—volunteers described how the measure was too time consuming for patients to complete, and that its length impacted on the conversation and rapport they were able to build with patients.[19] Furthermore, in recognition of the potential difficulties described above, researchers in Australia have produced a revised version of the measure for use with vulnerable older adult groups.[20]

## OBJECTIVES

The objectives of this research were to produce two measures that are practical, feasible and simple for hospital inpatients to complete. As it is currently conceived, the PMOS has a diagnostic function. It can be used to identify those factors (see domains above) where attention might most usefully be focused when planning improvement. The first objective therefore was to produce a revised version of PMOS that was shorter with improved acceptability, while at the same time preserving the psychometric properties, conceptual underpinnings and diagnostic function of the longer version. This version would continue to be used, alongside the PIRT tool as a basis for the PRASE intervention. Our second objective was to develop a version of PMOS that was much shorter and that could be used as a brief standalone measure for ongoing monitoring of ward safety performance not focused on planning improvement. Potentially, this measure could be used to prompt additional information gathering effort using the refined diagnostic measure or to monitor safety performance over time.

### Patient and public involvement statement
Patients and the public were involved in this study in the shortening and rewording of the questionnaires. More information is provided in the method.

## METHOD
First, we focused on the diagnostic measure and aimed to reduce the number of PMOS items, rephrase items where appropriate, consider face and content validity, and establish the internal reliability of the revised diagnostic measure. Next we focused on producing a brief standalone measure for on-going monitoring of ward safety performance and establishing its internal reliability, face validity, convergent validity and discriminant validity.

### Participants
#### PMOS-30
The shortening exercise included the analysis of PMOS data collected using the original measure (44 items) within the RCT of the PRASE intervention.[11] Analyses included 2002 patient participants recruited across three hospital trusts and 33 wards in the UK between May 2013 and September 2014. This exercise also included an iterative consensus approach in a series of meetings with researchers from the Yorkshire Quality and Safety Research (YQSR) Group who developed the original measure, and six researchers and research nurses with experience of facilitating the collection of PMOS data in the RCT of the PRASE intervention. These meetings aimed to reduce the number of items in the measure, and considered the phrasing of retained items. At this stage, to establish face validity, we also sought input on item wordings/phrasings from two patient panel representatives (from the YQSR Group's patient panel), and two hospital volunteers involved in a project implementing PRASE in collaboration with hospital volunteers, which henceforth refer to as the hospital volunteers project[19 21]

Subsequently, the internal reliability of the revised PMOS was tested in a sample of 751 patients. Patients were recruited by hospital volunteers across 22 wards from three hospitals, at three hospital trusts in the UK between September 2015 and May 2016, within the hospital volunteers project. The characteristics of the patient samples are presented in table 1.

#### PMOS-10
The shortening exercise to produce a brief standalone measure took a statistical approach, which involved

**Table 1** Characteristics of the patient samples

| | PMOS-44 N=2002 Data collected 2013–2014 | | PMOS-30 N=751 Data collected 2015–2016 | | PMOS-10 N=165 Data collected 2017–2018 | |
|---|---|---|---|---|---|---|
| Ward type | Medicine=41.20% Surgery=56.60% Mixed=2.20% | | Medicine=41.10% Surgery=49.00% Paediatrics=9.90% | | Medicine=49.70% Surgery=50.30% | |
| Age median, range | 65, 87 | | 60, 100 | | 57, 71 | |
| Gender | Female=50.0% Male=50.0% | | Female=53.5% Male=44.7% | | Female=55.2% Male=44.8% | |
| Length of stay in days median, range | 3, 167 | | 4, 90 | | 3, 49 | |
| Ethnicity | African | 0.3% | African | 0.1% | African | 1.2% |
| | Bangladeshi | 0.1% | Bangladeshi | 0.1% | Bangladeshi | 0.6% |
| | British | 93.5% | British | 89.7% | Black African | 0.6% |
| | Caribbean | 0.6% | Indian | 1.5% | Indian | 1.2% |
| | Chinese | 0.2% | Irish | 0.7% | Irish | 0.6% |
| | Indian | 0.5% | Other | 1.9% | Other Asian Background | 0.6% |
| | Irish | 0.6% | Other Background | 0.1% | Pakistani | 7.9% |
| | Other | 1.2% | Other Ethnic Background | 0.5% | White & Black Caribbean | 0.6% |
| | Other Background | 0.3% | Pakistani | 4.3% | White & Asian | 1.2% |
| | Other Mixed Background | 0.2% | White & Asian | 0.1% | White British | 83.6% |
| | Pakistani | 1.0% | Missing | 0.9% | Other White Background | 1.8% |
| | White & Black Caribbean | 0.2% | | | | |
| | Missing | 1.1% | | | | |

analyses of PMOS data collected using the original measure within the RCT of the PRASE intervention.[11] The characteristics of the patient samples are presented in table 1.

Prior to data collection, members of the YQSR Group's patient panel reviewed the measure resulting from the statistical exercise to ensure it was acceptable (face validity). The subsequent reliability and validity analyses included 165 patient participants who were recruited across nine wards in one hospital trust in the UK between June 2017 and January 2018. Patient eligibility criteria were those reported in the RCT of the PRASE intervention.[11]

### Recruitment and informed consent
#### PMOS-30
Patients were invited to complete the questionnaire by hospital volunteers within a quality improvement project; therefore, informed consent was not required. The hospital volunteer training package was based on the training for researchers and research nurses who facilitated patient data collection within the RCT of the PRASE intervention.[11] More information regarding the

procedure for hospital volunteers visiting wards has been published previously.[19]

#### PMOS-10
Patients were recruited by a research nurse and an undergraduate industrial placement student. Posters were placed on wards, and staff were encouraged to inform patients that researchers were recruiting. Researchers liaised with the nurse in charge to identify patients who had capacity and were considered well enough to take part. Researchers then approached patients to give them information about the study, both written (in the form of a participant information sheet) and verbally. If the patient agreed to take part and gave informed consent, they were recruited into the study.

### Measures
The response options for PMOS-44 (44 items), PMOS-30 (30 items) and PMOS-10 (10 items) are as follows: 1=strongly disagree, 2=disagree, 3=neither agree or disagree, 4=agree, 5=strongly agree, not applicable, and I prefer not to answer. Responses are recoded such that a high score indicates more favourable perceptions.

PMOS-44 and PMOS-30 questionnaires were completed using a laptop/handheld device. PMOS-10 questionnaires were collected on paper. For all versions of the PMOS questionnaire, patients could complete the questionnaire themselves, or completion could be facilitated (researchers/research nurses—PMOS-44; hospital volunteers—PMOS-30; research nurse/industrial placement student—PMOS-10). PMOS-44, PMOS-30, PMOS-10 and 'easy read' versions which have subsequently been produced can be requested via https://www.improvementacademy.org/tools-and-resources/patient-reporting-and-action-for-a-safe-environment.html (freely available).

To examine the convergent validity of PMOS-10, the Friends and Family Test (FFT)[22] question was asked at the beginning of the PMOS-10 questionnaire. The FFT is a one item measure of patient experience which asks if people would recommend the services they have used and offers a range of responses (extremely likely; likely; neither likely nor unlikely; unlikely; extremely unlikely)—'How likely are you to recommend this ward to friends and family if they needed similar care or treatment?' A lower score on this measure indicated more favourable perceptions.

### Analysis
All analyses were performed using SPSS V.23.[23] Our analytical approach was informed by the original PMOS validation study.[14] For PMOS data, if a patient responded 'not applicable' or 'I prefer not to answer' this was treated as missing data as these responses do not contribute to PMOS scoring. These two additional response options accounted for 87.39% of missing data points for PMOS-44, 72.39% for PMOS-30 and 99.44% for PMOS-10. The PMOS-30 and PMOS-10 shortening analyses only included PMOS-44 data where collection was facilitated by a researcher/research nurse, so we could be more confident in the reliability of responses. The PMOS-30 and PMOS-10 reliability and validity analyses included both facilitated and self-completed responses. All missing data were excluded list-wise, the characteristics of patients with missing data were consistent with the larger samples and the retained number of cases differed dependent on the type of analysis performed. There were the following numbers of complete PMOS responses across the data sets: PMOS-44 (n=1119); PMOS-30 (n=347); PMOS-10 (n=119), there was no missing data for the FFT question.

### Estimate of sample size
For the PMOS-30 and PMOS-10 reliability and validity analyses, the sample sizes were informed by both statistical and pragmatic considerations, based on a minimum subject to item ratio of at least 10:1.[24 25]

### PMOS-30
Researchers and research nurses who collected PMOS data from patients within the RCT of the PRASE intervention met to discuss the content of the original measure. Potential items for removal were highlighted, for example,

based on patient difficulties in understanding of the item or patient feedback that items were repetitive. Prior to a series of consensus meetings analyses were agreed, the results of which would be brought to the consensus meetings. These analyses included item response variation (Standard Deviation; SD) and inter-item correlations using Pearson's correlation.

Research suggests that for a 5-point scale items which demonstrate reasonable variability (SD of 1.00 or higher) are potential items to retain and that items with limited variance are candidates for removal.[26] On this basis, items demonstrating low variability (ie, <0.7) were identified as candidate items for removal, and in terms of the level of overlap between items (possible multicollinearity), high inter-item correlations highlighted candidate items for removal, that is, higher than 0.5.[27]

Subsequently, in a series of consensus meetings with the wider research group, items which had been identified as problematic, based on either researcher/research nurse perceptions or the item analyses, were discussed in greater detail. For the purpose of construct validity, the group considered each potential item for removal, and focused on whether each item reliably tapped what it was intended to measure. For the purpose of content validity, we also endeavoured to ensure a minimum of two items per domain were retained. We used multiple criteria, rather than only a statistical approach when considering items for removal to safeguard against key items being removed.

### Reliability analyses
We examined the internal consistency reliability (Cronbach's alpha) of PMOS-30 data collected from 751 hospital in-patients within the hospital volunteers project. In line with guidance, a reliability coefficient α of >0.7 was deemed acceptable.[28 29] For domains with three or more items, we examined the internal consistency reliability in terms of original measure's domain structure, that is, the domains that form the basis of the feedback to support service improvement in the PRASE intervention, and assessed the average inter-item correlations using the recommended values range of 0.15–0.50.[30] For two item domains, we assessed inter-item correlations.

### PMOS-10
Following a method described by Marteau and Bekker,[31] PMOS items were ranked in order of magnitude of their item-remainder correlations using Pearson's correlation (corrected item-total). PMOS-10 items were selected based on the highest corrected item-total correlations across both PMOS-44 and PMOS-30. PMOS-30 refers to items retained in the 30-item measure. These analyses were based on the PRASE RCT data; therefore, subsequent rewording and rephrasing of PMOS-30 items were not accounted for. We also examined how PMOS-10 correlated with PMOS-44 and PMOS-30.

**Table 2** Rationale for item removal

| Item removed | Rationale | |
|---|---|---|
| | **Consensus group discussion** | **Item analysis** |
| The following aspects of the ward made it difficult for staff to do their jobs: lighting levels | Researchers/research nurses highlighted that frequently patients did not respond to this item with regard to themselves, and therefore the item was often not answered as intended, that is, how lighting levels affect staff. | Low variability SD=0.59 |
| Staff were always able to get advice from other teams about my care if needed | Researchers/research nurses highlighted that frequently patients did not respond to this item in terms of what the item is attempting to tap, that is, specialist staff, etc. | Low variability SD=0.67 High inter-item correlations with 'I got answers to all the questions I had about my care' (0.51) |
| When staff talked about my care with others, the information they shared was correct | Researchers/research nurses highlighted that frequently patients did not respond to this item in terms of what the item is attempting to tap that is, information sharing between staff. | Low variability SD=0.61 |
| The following aspects of the ward made it difficult for staff to do their jobs: position of nurses' station | Researchers/research nurses highlighted that participants often struggled to respond to this item from the patient perspective. | Low variability SD=0.71 |
| There was equipment that staff found difficult to use (eg, monitoring equipment, beds, hoists) | Researchers/research nurses highlighted that frequently patients reported that it would be more appropriate for staff to respond to this item. | Low variability SD=0.64 |
| Staff were kept waiting for my test results | Researchers/research nurses felt this issue was covered elsewhere in the measure, that is, 'Information about me that my health care team needed was always available'. | |
| The following aspects of the ward made it uncomfortable for me: lack of space | Researchers/research nurses felt that lack of space was assessed elsewhere in the measure, although in relation to staff rather than patients. 'The following aspects of the ward made it difficult for staff to do their jobs: Lack of space'. | High inter-item correlations with 'The following aspects of the ward made it difficult for staff to do their jobs: Lack of space' (0.51) |
| Staff always seemed to know what they were meant to be doing | Researchers/research nurses highlighted that patients often felt they could not answer this. Researchers/researcher nurses also felt there was conceptual overlap with another item, that is, 'I got answers to all the questions I had about my care'. | Low variability SD=0.59 |
| Staff always agreed about my treatment/ care | Researchers/research nurses felt there was conceptual overlap with another item, that is, 'Staff gave me different information about my care'. | Low variability SD=0.60 |
| Nurses were always able to get help from other staff when they asked for it | Researchers/research nurses felt there was conceptual overlap with another item, that is, 'Staff seemed to struggle to get help when they needed it'. | Low variability SD=0.59 |
| I knew who to go to if I needed to ask a question | Researchers/research nurses felt other items capture this issue. | |
| The following aspects of the ward made it difficult for staff to do their jobs: clutter and untidiness | | Low variability SD=0.67 |
| Equipment needed for my care was always working properly | Researchers/research nurses highlighted that frequently patients reported that it would be more appropriate for staff to respond to this item. | Low variability SD=0.71 |
| The following aspects of the ward made it uncomfortable for me: other—please specify | Researchers/research nurses felt this item could be removed due to low response rate. | |

SD, Standard Deviation.

## Validity and reliability analyses

Taking the same approach as the original PMOS validation study, an overall PMOS-10 'positive index' was constructed by summing the number of items that patients responded to by using one of the two positive response options (strongly agree or agree) for a positively worded item and (strongly disagree or disagree) for a negatively worded item. Therefore, patients could have a score of 10 where higher responses equate to more favourable safety perceptions. If patients had missing responses, the 'positive index' was constructed from the available data.

## Internal reliability

We assessed the internal consistency reliability of PMOS-10 (Cronbach's alpha), and in line with guidance, a reliability coefficient α of >0.7 was deemed acceptable.[28 29]

**Table 3** Retained items rephrasing

| PMOS-44 item | PMOS-30 item |
| --- | --- |
| Sometimes there was no one available to deal with aspects of my care | There was always someone available to deal with every aspect of my care |
| Staff did not work together as a team here | Staff worked together as a team here |
| The drugs I have been prescribed were always available in the hospital | My medicines were always available |
| A doctor changed my plan of care and other staff did not know about it | Staff did not always know when a doctor changed my plan of care |
| Staff gave me different information about my care | Staff gave me conflicting information about my care |
| I have needed treatment and there was no one available who was trained to do it | When I needed treatment, there was always someone available who was trained to do it |
| Equipment and supplies were not always available when needed (eg, hoists, bed pans, drugs) | Equipment and supplies were always available when needed (eg, hoists, bed pans, walking aids, dressings) |
| I was on a ward that was not able to deal with my treatment needs | The ward was able to deal with all my treatment needs |
| After a shift change, staff did not appear to know important information about my care | After shift changes, staff knew important information about my care |
| It was clear who was in charge of the staff | It was clear who was in charge of the ward staff |
| I always knew which nurse was responsible for my care | I always knew which nurse or nurses were responsible for my care |
| On at least one occasion, a member of staff was not able to use the necessary equipment | Staff were always able to use the necessary equipment |
| On at least one occasion, a member of staff was not able to carry out a task that they should have been able to do | Staff were always able to carry out tasks that they should be able to do |
| My treatment/ procedure/operation did not always happen on time | My treatment/procedure/operation always happened on time |

## Convergent validity

To determine the convergent validity of the 10-item questionnaire (ie, whether the measure converges with an existing related measure), we examined the association between the PMOS-10 'positive index' and the FFT scores using Pearson's correlation. In line with Cohen's guidelines,[32] correlations of 0.1 were interpreted as a small effect, 0.3 as medium and 0.5 as large.

## Discriminant validity

A one-way analysis of variance (ANOVA) was conducted to assess the extent to which PMOS-10 discriminated among the nine wards. Ward was entered as a fixed factor, and the PMOS-10 'positive index' as the dependent factor.

## RESULTS
## PMOS-30

Of the 44 items in the original measure, 14 items were removed (32%). Table 2 presents the rationale for the removal of each item. Eight items were removed due to both statistical reasons and researcher/research nurse feedback, five items were removed due to researcher/research nurse feedback alone and one item was removed due to statistical reasons alone. To facilitate the likelihood of reliable and valid responses, we also reduced the quantity of negatively phrased items in the revised measure. In the original measure, the proportion of negatively phrased items was 57% (25 items); in the revised measure,

this was reduced to 33% (10 items). See table 3 or the rephrasing of retained items.

## Internal reliability

The revised measure (PMOS-30) demonstrated strong reliability, with a coefficient α of 0.89. Cronbach's alphas and average inter-item correlations were calculated for domains with three or more items, and inter-item correlations (Pearson) were calculated for domains with two items (table 4). Cronbach's alphas for five of the domains ranged from 0.40 to 0.75, and for these domains, the average inter-item correlations (Pearson) were all within an acceptable range (0.17–0.38), as were the inter-item correlations (Pearson) for the two item domains (0.23–0.42).

## PMOS-10

Table 5 shows the 10 highest items (based on corrected item-total correlation) that were included in both PMOS-44 and PMOS-30. These items formed PMOS-10. PMOS-10 correlated with the PMOS-44 (r=0.90, p=0.000) and PMOS-30 (r=0.91, p=0.000).

The mean PMOS-10 'positive index' index score for the entire sample was 8.37 (SD=1.65), meaning that on average, patients responded positively to around eight of

 Louch G, et al. BMJ Open 2019;9:e031355. doi:10.1136/bmjopen-2019-031355

**Table 4** PMOS-30 internal reliability domain scores

| Domain, number of items | Items | Cronbach's alpha | Average inter-item correlation* |
|---|---|---|---|
| Dignity and respect (not linked to any domains), 1 | I was always treated with dignity and respect | N/A | N/A |
| Domain 1. Communication and teamworking, 5 | I got answers to all the questions I had about my care | .753 | .383 |
| | I always felt staff listened to me about my concerns | | |
| | There was always someone available to deal with every aspect of my care | | |
| | I felt that the attitude of staff towards me was poor (R) | | |
| | Staff worked together as a team here | | |
| Domain 2. Organisation and care planning, 4 | My medicines were always available | .521 | .220 |
| | Staff didn't always know when a doctor changed my plan of care (R) | | |
| | Staff gave me conflicting information about my care (R) | | |
| | When I needed treatment there was always someone available who was trained to do it | | |
| Domain 3. Access to resources, 3 | Staff/patients waited a long time for porters to arrive (R) | .401 | .183 |
| | Staff seemed to struggle to get help when they needed it (R) | | |
| | Equipment and supplies were always available when needed (eg, hoists, bed pans, walking aids, dressings) | | |
| Domain 4. Ward type and layout, 7 | Staff were prompt in answering my buzzer | .589 | .171 |
| | The ward was able to deal with all my treatment needs | | |
| | Lack of space made it difficult for staff to do their jobs (R) | | |
| | The following aspects of the ward made it uncomfortable for me: Noise levels (R) | | |
| | The following aspects of the ward made it uncomfortable for me: Lighting levels (R) | | |
| | The following aspects of the ward made it uncomfortable for me: Temperature (R) | | |
| | The following aspects of the ward made it uncomfortable for me: Poor cleanliness (R) | | |
| Domain 5. Information flow, 2 | Information about me that my healthcare team needed was always available (eg, drug charts, medical notes, test results) | N/A | .341* |
| | After shift changes staff knew important information about my care | | |
| Domain 6. Staff roles and responsibilities, 4 | I knew what the different roles of the people caring for me were | .613 | .292 |
| | It was clear who was in charge of the ward staff | | |
| | I knew which consultant was in charge of my care | | |
| | I always knew which nurse or nurses were responsible for my care | | |
| Domain 7. Staff training, 2 | Staff were always able to use the necessary equipment | N/A | .423* |
| | Staff were always able to carry out tasks that they should be able to do | | |

Continued

| Table 4 | Continued | | |
|---|---|---|---|
| Domain 8. Delays, 2 | There were enough staff on the ward to get things done on time | N/A | .238* |
| | My treatment/ procedure/ operation always happened on time | | |

Notes: (R) indicates item to be reverse coded so higher scores=safer care.

*Inter-item correlation reported for domains with two items.

N/A, not applicable.

the PMOS-10 items. The recruitment response rate was 92%.

### Internal reliability

The PMOS-10 measure demonstrated good reliability, with a coefficient α of 0.79.

### Convergent validity

A significant negative correlation was established between the PMOS-10 'positive index' and the FFT scores (r=−0.36, p=0.000), indicating that the more positive PMOS scores among patients, the more positive FFT scores were.

### Discriminant validity

A one-way between-groups ANOVA was performed to assess the extent to which the PMOS positive index discriminated among the nine wards. There was no significant main effect of ward ($F (8, 156)=1.59$, p=0.130).

### DISCUSSION

We aimed to produce two revised patient measures of safety that are practical, feasible and simple for patients to complete, responding directly to the need to enhance patient acceptability of the original measure.[19 20] PMOS-30 included less negatively worded items compared with PMOS-44 (37% as opposed to 57%), and the internal reliability of PMOS-30 was established. Reducing the number of negatively worded items addressed the potential for unreliable responses, as previous research suggests that reverse worded items can be contaminated by inattention and confusion.[33] We retained a minimum of two items per domain across the nine domains known to contribute to safety incidents in hospitals and which formed the basis of the original measure,[13 14] informed by the YCFF.[15] In addition, and in collaboration with our patient representatives, items were rephrased to facilitate greater patient understanding. One domain demonstrated an acceptable Cronbach's alpha coefficient and acceptable average inter-item correlation: communication and teamworking. Four domains did not demonstrate an acceptable Cronbach's alpha coefficient but demonstrated an acceptable average inter-item correlation: organisation and care planning, access to resources, ward type and layout and staff roles and responsibilities. Three domains (two item domains) demonstrated acceptable inter-item correlations: information flow, staff training and delays. The reduction in the number of items may have resulted in lowering the reliability statistics of the domains; however, this is to be expected as a reliability coefficient generally increases as the number of items increases.[29] Interestingly, the domains which performed poorest (eg, ward type and layout; access to resources) also performed poorest in the original PMOS validation study.[14] Collectively, these findings demonstrate the stability of the original domains over time, even within a shorter, revised measure and suggests that the conceptual underpinnings and psychometric properties of the original measure have been preserved.

**Table 5** Retained items for PMOS-10

| Original wording of item in PMOS-44 | Item wording in PMOS-30 | Domain (PMOS-44 and PMOS-30) | PMOS-44 corrected item-total correlation | PMOS-30 corrected item-total correlation |
|---|---|---|---|---|
| I got answers to all the questions I had about my care | I got answers to all the questions I had about my care | Communication and teamworking | 0.58 | 0.56 |
| A doctor changed my plan of care and other staff did not know about it | Staff did not always know when a doctor changed my plan of care | Organisation and care planning | 0.50 | 0.48 |
| After a shift change, staff did not appear to know important information about my care | After shift changes, staff knew important information about my care | Information flow | 0.52 | 0.49 |
| Staff were prompt in answering my buzzer | Staff were prompt in answering my buzzer | Ward type and layout | 0.49 | 0.51 |
| Sometimes there was no one available to deal with aspects of my care | There was always someone available to deal with every aspect of my care | Communication and teamworking | 0.59 | 0.59 |
| On at least one occasion, a member of staff was not able to carry out a task that they should have been able to do | Staff were always able to carry out tasks that they should be able to do | Staff training | 0.54 | 0.52 |
| I felt that the attitude of staff towards me was poor | I felt that the attitude of staff towards me was poor | Communication and teamworking | 0.50 | 0.50 |
| I have needed treatment and there was no one available who was trained to do it | When I needed treatment, there was always someone available who was trained to do it | Organisation and care planning | 0.55 | 0.53 |
| I always felt staff listened to me about my concerns | I always felt staff listened to me about my concerns | Communication and teamworking | 0.56 | 0.53 |
| Staff seemed to struggle to get help when they needed it | Staff seemed to struggle to get help when they needed it | Access to resources | 0.51 | 0.48 |

The 10 most highly correlated items that were included in both PMOS-44 (original measure) and PMOS-30 formed the briefer standalone measure (PMOS-10), which was ratified in collaboration with patient representatives to establish face validity. A good alpha coefficient demonstrated the internal reliability of PMOS-10, and convergent validity was established. The strength of the correlation (moderate) suggests that although PMOS-10 'converges' with another known measure of care experience from the patient perspective (FFT), it is indeed measuring something different. To determine discriminant validity, we examined whether PMOS-10 was able to discriminate meaningfully at the ward level, with a view to PMOS-10 being used as a tool to identify those wards which might benefit most from further data gathering efforts (ie, PMOS-30) and improvement as part of the PRASE intervention. There was no significant main effect of ward and no significant differences between wards. Therefore, based on our analyses of nine wards, we cannot purport that PMOS-10 is able to meaningfully discriminate between wards.

There has been substantial research and policy focus on patient involvement in patient safety.[34–37] Patients providing feedback about the safety of their care is one way for patients to be engaged in patient safety.[38] The collection of feedback on patient satisfaction and experiences of care are also commonplace.[4] In terms of safety specifically, there is a wealth of research in support of patients providing feedback about the safety of their care,[8 9] and the PRASE intervention, is the first of its kind to collect patient feedback about safety on an on-going basis to support service improvement. Vincent and colleagues[3] emphasised that patients and carers play an essential role in safety monitoring, but are often an underused resource. To move from patients and carers being an underused resource in the monitoring of safety, we need to ensure that the measures we do have to capture their perspective are acceptable to patients and carers, are not burdensome to complete and are reliable and valid. The research presented in this paper responds directly to these needs.

### Strengths and limitations

A strength of the development of PMOS-30 is the involvement of six researchers/research nurses with an extensive amount of experience of facilitating PMOS (original measure) data collection, and the involvement of two hospital volunteers and two patient representatives. This provided a rich source of data regarding potential rephrasing of items; leading to considerable face validity. A further strength is that the reliability testing of PMOS-30 took place in a variety of wards across three hospital trusts. The involvement of two patient representatives in the development of PMOS-10 is a key strength.

A limitation of our work is that our studies only included patient participants who could understand English. Therefore, further testing would be required if PMOS-30 and PMOS-10 were to be translated into other languages. For PMOS-30, some of the domains failing to achieve Cronbach's alpha recommendation of 0.70 is a limitation, even though the average inter-item and inter-item correlations were within the optimum range. For PMOS-10, although the wards recruited were from a variety of medical and surgical specialities, a limitation is the involvement of only one hospital trust. A further limitation is that discriminant validity was not established for PMOS-10. Finally, PMOS-30 had a high amount of missing data which were mainly not applicable responses; therefore, it is possible that a listwise deletion approach may have implications for generalisation.

### Implications for health services research, policy and practice

By further refining an existing measure, the findings respond to previous research which called for the acceptability of PMOS to be enhanced. The length and some content of the original measure were problematic, and posed issues relating to participant fatigue and burden. The output of a highly reliable and valid shortened diagnostic version of PMOS, which retains the psychometric properties and conceptual underpinnings of the original measure, is integral to the wider adoption and ongoing implementation of the PRASE intervention. The high recruitment response rate (92%) for PMOS-10 strengthens the argument that the measure is acceptable to patients, and these findings position PMOS-10 as a reliable and valid brief standalone measure for ongoing monitoring of ward safety performance from the patient perspective. To further validate PMOS-10, research is needed to assess the discriminant validity PMOS-10 on larger scale, in addition to further exploring convergent validity by examining associations with other indicators of patient safety.

### CONCLUSION

Two revised, shortened versions of the original PMOS-44 (PMOS-30 and PMOS-10) were produced to capture patient feedback about safety in hospital, and these measures demonstrated acceptable reliability and validity. PMOS-30 was produced to serve a diagnostic function to be used, alongside the PRASE intervention as a whole. PMOS-10 was developed to be used as a brief standalone measure for ongoing monitoring of ward safety performance.

**Acknowledgements** We would like to thank the Yorkshire Quality and Safety Research (YQSR) Group patient panel for reviewing the PMOS-10 measure, and we would like to thank the two members of the panel and two hospital volunteers for their time and input into revising the PMOS-30 measure. We would also like to thank the patient participants and the staff on the wards for their contribution to the research.

**Collaborators** Yorkshire Quality and Safety Research Group

**Contributors** GL led on the development of PMOS-30, carried out the related analyses and drafted the manuscript. JH carried out the PMOS-10 shortening statistical analyses. CR led on the PMOS-10 data collection, and GL, AA and CR carried out the related analyses. SM and CM supported the studies at all stages, in particular the patient and public involvement and engagement element of this work. RL provided guidance and input at all stages. All authors provided comments and approved the final version.

**Funding** The research was supported by The Health Foundation, the National Institute for Health Research (NIHR) Yorkshire and Humber Patient Safety Translational Research Centre (NIHR YH PSTRC) and NIHR CLAHRC Yorkshire and Humber www.clahrc-yh.nihr.ac.uk. The PMOS-30 and PMOS-10 shortening exercises utilised data from a wider study funded by the NIHR under its programme Grants for Applied Research scheme ('Improving patient safety through the involvement of patients', RP-PG-0108-10049). The views expressed in this article are those of the author(s) and not necessarily those of The Health Foundation, the NHS, the NIHR or the Department of Health and Social Care.

**Competing interests** None declared.

**Patient consent for publication** Not required.

**Ethics approval** Ethical approvals were obtained from the University of Bradford, Humanities, Social and Health Sciences Research Ethics Panel, and from the East Midlands - Leicester Central Research Ethics Committee.

**Provenance and peer review** Not commissioned; externally peer reviewed.

**Data availability statement** The data that support the findings of this study are available from the corresponding author, GL, upon reasonable request. PMOS-44, PMOS-30, PMOS-10 and 'easy read' versions which have subsequently been produced can be requested via https://www.improvementacademy.org/tools-and-resources/patient-reporting-and-action-for-a-safe-environment.html (freely available).

**ORCID iDs**
Gemma Louch http://orcid.org/0000-0001-6946-3693
Rebecca Lawton http://orcid.org/0000-0002-5832-402X

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
