## [Reviewer comments · BMJ Open]

ARTICLE DETAILS

TITLE (PROVISIONAL)	Validation of revised Patient Measures of Safety: PMOS-30 and PMOS-10
AUTHORS	Louch, Gemma; Reynolds, Caroline; Moore, Sally; Marsh, Claire; Heyhoe, Jane; Albutt, Abigail; Lawton, Rebecca

VERSION 1 – REVIEW

REVIEWER	Dr. M. A. van Melle Research associate University of Cambridge/ NHS Cambridge & Peterborough Trust UK
REVIEW RETURNED	02-Jun-2019

GENERAL COMMENTS	Thank you for giving me the opportunity to review this interesting article. I consider it an important aim. Patient input is invaluable in patient safety improvement. I think the change in wording (Table 3) have improved the items a lot. It is not easy writing readable questions and I'm glad the authors had patient input in that. To improve the readability, I would like to suggest some structural changes with consistency improvements and would like to request more information on the methods of the study. Major revisions: 1. The Introduction is very long. It paints a good picture, but many parts are redundant, especially the referral to the previous study (paragraph 2 not needed, 4 & 5 could be merged and summarised). Making the introduction shorter will leave more words for the methods section; to provide more information and getting rid of the footnotes. For example, the methods section for phase II is very short and incomplete and the information in the footnotes should be in the main text.2. The first paragraph of the discussion is also largely repeating the results section; this can be summarised to give more space for adding more information to the methods section.3. The structure of the article is somewhat confusing; phase I and II are almost separate articles as if written by a different author; with their own subheadings, methods and results
---

sections-impeding the readability of the text. It would be advisable if the authors could first address all of the methods (also limiting double information) for both the phases and then the results. And given that both a Stage 1 and Stage 2 methods are largely the same (except the addition of convergent and discriminant validity in the PMOS 10) I would leave out the 'Stage 1' and 'Stage 2' and just give them clear subheadings for the shortening exercise and construct validity. Or give subheadings for the names of the questionnaires instead of Phase 1 and 2 (PMOS-30 and PMOS-10).

I would be grateful if the authors could add some consistency in the subheadings; both phases are currently structured differently; for instance the PMOS 30 would also benefit from a recruitment and informed consent, sample size and measures paragraph.
Btw: Makes you wonder why for the PMOS 30 no construct validity (convergent and discriminant validity) was assessed.

4. The authors state that this research is performed in a sample of 2002 patients (in the bullet points and the discussion section). However, I beg to differ this number in the strengths of the article (in the bullet points and in the discussion section). It refers to the previous research that has been done. As I understand from the article, the authors have taken the previous results, have discussed all the items in the PMOS 44 and reassessed these items to make a shorter questionnaire by consensus (so not asking or using these 2002 patients) Afterwards, they have a new population of 465 (and 165 for the PMOS 10). So I believe the population for this current study is 425 and 165.

5. Additional methods information:

- Could you shortly describe the PMOS 30 and PMOS 10 after the shortening exercise or in the descriptive statistics; the domains and numbers in it. Was a 5-point scale used?
- Could you describe the means of the patients filling in the questionnaire; did they do it online, did they have help? The fieldworkers mentioned in the Table 2; when were they used? Are means of filling in the same in all the PMOS versions?
Is this the same in the FFT questionnaire?
- The discussion refers to the data collection method, however this is not described in the methods section.
- The description of field worker input is useful (however as it is in the results, please also describe the means in the methods; I see it is mentioned in the introduction).
- The % missings is not mentioned, which is also very important because any patient with at least one missing item was excluded because of the complete case

	analysis: How many and what type of patients were excluded? Could you describe the distribution per ward shortly in the methods and Table 1 and 5? Patients on different wards can be widely different (for instance Internal/diagnostic medicine vs Surgery) and it would help to understand the type of patients included and could influence validity of the questionnaire.  - Table 6 gives two different wordings: which wording was used in the validation of the PMOS10? Why present both? Which group of patients was used for these values? The 165 new patients? Surprising to see almost all the communication and teamworking items have been used in PMOS 10 (40% of the entire short version; although I consider this a very important part of patient safety!). I would suggest to presented the PMOS 44/30 dimension items together in the table? 6. I would discuss whether the ward type and layout dimension items in the PMOS are valid in one dimension. The name of the dimension seems not to fit and the items seem to be two separate dimensions. One could beg to differ whether the staff promptness and treatment needs would fit a name such as ward type and layout; or is it access? This is portrayed by the low inter-item correlation as well. Maybe consider a change of dimension name or would a new EFA or SEM be of value; domain 3 also has problems and seems related to domain 4 (at least at face value). 7. Suggested additions to the discussion section  - Could the author elaborate on the cause of poor cronbach's alpha and/or inter-item correlation for some of the dimensions? What would be a possible cause? - I cannot completely review the limitation section, because I am missing methods information (such as recruitment on wards and response) Minor revisions:  1. Abstract: would like to see the numbers for the reliability statements in the results 2. PMOS covers In-hospital care-> please describe somewhere in aim. 3. Table 1: the gender numbers do not add up to 100% Are all these others excluded? The other 2% missing? 4. The discussion gives the recruitment rate, however this is not mentioned in the results. Why not? Please add response/recruitment rate to the results section of both questionnaires.
--	--

	5. Typos: p18, line 25 -> two 'to's' to to enhance P18 line 50-> two 'of's' 6. References: there is definitely more recent material then Vincent 2002; a lot has happened in patient safety since then. 7. I do not see a checklist (COSMIN would be a relevant one; although only relevant parts) 8. I cannot find any funding information
--	--

REVIEWER	Solvejg Kristensen Aalborg University Hospital - Psychiatry Mølleparkvej 10 9000 Aalborg Denmark
REVIEW RETURNED	21-Jun-2019

GENERAL COMMENTS	Thank you for the opportunity to review this manuscript. I found the paper engaging and the research undertaken to be contemporary and topical. It is a most interesting and needed research which is highly relevant to the international audience. The paper aims to produce two revised, shortened versions of the original 44-item Patient Measure of Safety (PMOS-44), which retained the psychometric properties of the PMOS-44. The paper is well written and well documented, it has a good logical flow in terms of content and although a bit differently reported than usual, the flow matches the iterative interactive mixed method research applied. The thorough and inclusive in terms of co-design and cooperation between volunteers, researchers, health care professionals and patients is a most welcome scientific contribution. On the overall, the research is well carried, but I do have quite a number of specific observations/comments/questions, which can be used to enhance clarity of the paper and ease the reader. 1. P 5, Objectives, line 27-35. Formulation of this section so that it matches the headline (Objectives) would be welcome, as it stands now it can be perceived as methods. And in this form it seems more appropriate as meta text at the beginning of the Methods section, where an introductions to "what kind of study was carried out" is missing momentarily 2. P 5 onwards, Methods section. A description of the PMOS-44 is missing, how many items/domains, how many negatively worded items? what kind of answer categories, how is the scores and result calculations? 3. P 6, Table 1 – is it possible to add the year of data collection? This would clarify 4. P 6, Analysis onwards. I would welcome a clearer description of the criteria set prior to the analysis for evaluation psychometric properties of the PMOS-30 and PMOS-10. This would ease the reader and give a firmer and more trustworthy base for the evaluation and conclusions. E.g. P 7 line 30: "We used multiple criteria, rather than only a statistical approach..." Please clarify both types of criteria somehow. 5. P 14, Methods, Participants, Stage 1. I miss mentioned the number of participants, not only a reference to table 1 6. P 14, Methods, Participants, Stage 2. In this section you mention 165 patients recruited across nine wards. A description of the setting
--

	seems needed in general 7. P 15, Measures. PMOS-10 This section seems to fall apart, is some text missing? It is not clear why the FFT is presented, it becomes clear later (P 20, line 16-18), but then actual technical information (number of items, domains, answer categories, scoring etc. is missing) 8. P 18, Discussion. Line 42, “Five domains.....” This sentence and the further needs clarification, e.g. Cronbach’s alpha was only calculated for 2 of the five domains, due to the item numbers in the domains. 9. P 19, Strengths and limitations; “there was also a number of researcher and hospital volunteers with an extensive amount of...” if this is highlighted as a strength, it is needed to know the volume of “a number of researcher and hospital volunteers”. There seems to be only one limitation in this study, usually survey-based studies have a number of limitations. Given these have been overcome, it might be worth mentioning for others to learn from. The implications of the strength and limitations are not clear to the reader, please specify. 10. Last comment, I do not find the supplementary reporting (e.g. trial registration; funding details; CONSORT, STROBE or PRISMA checklist)
--	--

VERSION 1 – AUTHOR RESPONSE

Reviewer 1	
The Introduction is very long. It paints a good picture, but many parts are redundant, especially the referral to the previous study (paragraph 2 not needed, 4 & 5 could be merged and summarised). Making the introduction shorter will leave more words for the methods section; to provide more information and getting rid of the footnotes. For example, the methods section for phase II is very short and incomplete and the information in the footnotes should be in the main text.	Thank you for this feedback. We have amended the introduction as suggested and moved information from the footnotes into the methods main text.
The first paragraph of the discussion is also largely repeating the results section; this can be summarised to give more space for adding more information to the methods section.	This has been amended as suggested.
The structure of the article is somewhat confusing; phase I and II are almost separate articles as if written by a different author; with their own subheadings, methods and results sections-impeding the readability of the text. It would be advisable if the authors could first address all of the methods (also limiting double information) for both the phases and then the results. And given that both a Stage 1 and Stage 2 methods are largely the same (except the addition of	Thank you for raising this. We did originally intend to present the development of PMOS-30 and PMOS-10 separately. However, after much deliberation within the research team we decided it was more appropriate to present both measures in one paper. We have attempted to limit duplicated information, have added the information suggested and have significantly re-formatted the structure of the paper. In 2014, the wider PRASE intervention (including PMOS tool) was implemented as part of a quality improvement

convergent and discriminant validity in the PMOS 10) I would leave out the 'Stage 1' and 'Stage 2' and just give them clear subheadings for the shortening exercise and construct validity. Or give subheadings for the names of the questionnaires instead of Phase 1 and 2 (PMOS-30 and PMOS-10). I would be grateful if the authors could add some consistency in the subheadings; both phases are currently structured differently; for instance the PMOS 30 would also benefit from a recruitment and informed consent, sample size and measures paragraph. Btw: Makes you wonder why for the PMOS 30 no construct validity (convergent and discriminant validity) was assessed.	project with hospital volunteers facilitating PMOS data collection. Running alongside the quality improvement project was a research evaluation. It was highlighted very early on in the research evaluation that the length of the PMOS questionnaire (44 items) and the phrasing of many of the questions were a threat to hospital volunteers remaining in the project. The hospital volunteers felt the measure was too long and impacted on the rapport they were able to build with patients. This triggered the need to reduce and amend the 44 item PMOS more immediately. The 751 (465 in previous version of manuscript) participants referred to represents data from the quality improvement project i.e. not primary research data. As this project was already underway across numerous wards and organisations, there was no scope to add in additional data to be collected to assess convergent validity. Furthermore, as the PMOS-30 data was not primary research data, there was not an even spread of responses across the wards, as was the case for the PMOS-10 data where the target number of 25 responses per ward was planned in advance of data collection.
The authors state that this research is performed in a sample of 2002 patients (in the bullet points and the discussion section). However, I beg to differ this number in the strengths of the article (in the bullet points and in the discussion section). It refers to the previous research that has been done. As I understand from the article, the authors have taken the previous results, have discussed all the items in the PMOS 44 and reassessed these items to make a shorter questionnaire by consensus (so not asking or using these 2002 patients) Afterwards, they have a new population of 465 (and 165 for the PMOS 10). So I believe the population for this current study is 425 and 165)	We agree and have removed this from the bullet points and the strengths section of the discussion.
Could you shortly describe the PMOS 30 and PMOS 10 after the shortening exercise or in the descriptive statistics; the domains and numbers in it. Was a 5-point scale used?	We have now included in the methods the response options for the PMOS measures, which were all on a 5-point likert scale with the additional options of 'not applicable' and 'I prefer not to answer'. A descriptive overview of PMOS-30 is provided which includes the domain and number of items in each domain (presented in Table 3). For PMOS-10 this information is presented in Table 5.
Could you describe the means of the patients filling in the questionnaire; did they do it online, did they have help? The fieldworkers mentioned in the Table 2; when were they used? Are means of filling in the same in all the PMOS versions? Is this the same in the FFT questionnaire?	Thank you for the opportunity to expand on this. PMOS-44 and PMOS-30 questionnaires were completed using a laptop computer or a handheld device. PMOS-10 questionnaires were collected on paper.  • For PMOS-44 patients could self-complete or a researcher/research nurse could facilitate data collection and read out and record responses. • For PMOS-30 patients or a family member/carer

	could self-complete or hospital volunteer could facilitate data collection and read out and record responses to the patients or family member/carer.  • For PMOS-10 patients could self-complete or a research nurse or industrial placement student could facilitate data collection and read out and record responses. For the PMOS-30 and PMOS-10 shortening exercises which used data from the PRASE RCT—following advice from a statistician we only included data in these analyses which was facilitated by a researcher/research nurse (n = 2002). The rationale for this was that we could be more confident in the reliability of responses when data collection was facilitated as there was less chance of unreliable responses as a result of negatively worded items being misunderstood. For the PMOS-30 and PMOS-10 validation the findings reported include self-completed and facilitated responses. In this previous version of the manuscript for the PMOS-30 validation we only include facilitated responses, however, we have amended this in the current version to include all responses (self-completed and facilitated) as we feel this is more appropriate. This means we now report 751 participants as opposed to 425. More widely than the data collection referred to in this manuscript, the use of the tools that comprise the PRASE intervention have become more widespread, and participating organisations choose the means of collecting data, which for PMOS-30 may be via the PRASE app (developed within the hospital volunteers project) on hand-held devices, or on paper. Currently, PMOS-10 is only collected on paper; however this may change over time. The NHS Friends and Family Test (FFT) is a feedback collection system for providers of NHS funded services. Providers are allowed to choose their collection mode, such as postcards, websites, phone calls, SMS etc, and are encouraged to use a range of collection modes. In the PMOS-10 validation study, the FFT question was asked alongside the PMOS questions.
The discussion refers to the data collection method, however this is not described in the methods section.	We have corrected this.
The description of field worker input is useful (however as it is in the results, please also describe the means in the methods; I see it is mentioned in the introduction).	The fieldworkers refer to the researchers and research nurses who facilitated PMOS-44 data collection in PRASE RCT. Instead of using the term fieldworkers, we now just refer to researcher and research nurses to avoid any confusion, and now refer to this in the method also.
The % missings is not mentioned, which is also very important because any patient with at least one missing item was excluded because of the complete case analysis: How many and what type of patients were	For all PMOS measures (44, 30 and 10), in addition to the 5-point likert response options, participants could also respond 'not applicable' and 'I prefer not to answer'. In all analyses these two additional response options were treated as missing data as they do not form part of the likert scale scoring.

excluded?	In the PMOS-44 dataset (used for the PMOS-30 and PMOS-10 shortening analyses) there were 2577 data points treated as missing, which are broken down as follows:  - not applicable (2194) 85.14% - I prefer not to answer (58) 2.25% - system missing (325) 12.61% In the PMOS-30 dataset there were 1492 data points treated as missing, which are broken down as follows:  - not applicable (1041) 67.77% - I prefer not to answer (69) 4.62% - system missing (382) 25.60% In the PMOS-10 dataset there were 52 data points treated as missing, which are broken down as follows:  - not applicable (48) 92.30% - I prefer not to answer (3) 7.14% - system missing (1) 1.92% Therefore, in all datasets the majority of data treated as 'missing' were instances where participants had chosen the 'not applicable' or 'I prefer not to answer' options, as opposed to system missing. To further explore missing data in terms of the descriptive information presented in Table 1 we created new datasets that only included participants with one or more missing data point. The characteristics of the sub-samples were consistent with the larger samples. We have incorporated the information presented in this response into the revision. NB - Exploring missing data meant going back to the raw PMOS-10 data which highlighted data input errors for seven participants, which have now been corrected. We re-ran all relevant analyses following the changes, and the ANOVA for discriminant validity main effect is no longer significant. We have amended this.
Could you describe the distribution per ward shortly in the methods and Table 1 and 5? Patients on different wards can be widely different (for instance Internal/diagnostic medicine vs Surgery) and it would help to understand the type of patients included and could influence validity of the questionnaire.	We now include the distribution of PMOS responses by ward type in Table 1. We have included this information at the level of medicine and surgery and paediatrics, as it is at that level we had the information across all three data sets.
Table 6 gives two different wordings: which wording was used in the validation of the PMOS10? Why present both? Which group of patients was used for these values? The 165 new patients?	Thank you for the opportunity to clarify this. This is to demonstrate the wording changes between PMOS-44 and PMOS-30. This is appropriate as the data analyses contributing to the development of PMOS-10 used the PRASE RCT PMOS-44 dataset and therefore this data used the original wording (data collection May 2013 to September 2014). The PMOS-30 data (data collection September 2015 to May 2016) also contributed to development of PMOS-10 and therefore we present that wording too.

	The PMOS-10 validation study (N=165) (data collection June 2017 to January 2018) utilised the most recent wording at that time, which was the PMOS-30 wording. This is explained in footnote i.
Surprising to see almost all the communication and teamworking items have been used in PMOS 10 (40% of the entire short version; although I consider this a very important part of patient safety!). I would suggest to presented the PMOS 44/30 dimension items together in the table?	Indeed, this is an important point and links to the rationale for the intended use of PMOS-10, which was to produce a brief standalone measure for on-going monitoring of ward safety performance. This warranted the purely statistical approach to highlight important items to retain. The domains items sit within the measure were the same for PMOS-44 and PMOS-30. We did not amend the domain labels or the items within domains as part of the development of PMOS-30. The rationale for this is covered in the next response. So this is clear in the paper, in Table 5 we have changed the column heading 'Domain (Original)' to 'Domain (PMOS-44 and PMOS-30)'.
I would discuss whether the ward type and layout dimension items in the PMOS are valid in one dimension. The name of the dimension seems not to fit and the items seem to be two separate dimensions. One could beg to differ whether the staff promptness and treatment needs would fit a name such as ward type and layout; or is it access? This is portrayed by the low inter-item correlation as well. Maybe consider a change of dimension name or would a new EFA or SEM be of value; domain 3 also has problems and seems related to domain 4 (at least at face value).	Thank you for highlighting this. For PMOS-30 we decided to present the items under the original domain headings and not to conduct a factor analysis to identify revised/new domains. We decided on this approach because PMOS-30 has been in use for over 4 years with a diagnostic function (as part of the PRASE intervention) across multiple organisations and wards in the UK. During this time, the PMOS-30 item responses were (and still are) presented to staff in a standardised feedback report which is linked around the original domains, and ward staff action plan based on the feedback report. Therefore, we feel it would be confusing to amend the domains at this point.
Discussion	
Could the author elaborate on the cause of poor cronbach's alpha and/or interitem correlation for some of the dimensions? What would be a possible cause?	Thank you for highlighting this, we now expand on this in the discussion.
I cannot completely review the limitation section, because I am missing methods information (such as recruitment on wards and response)	We have provided more methods information so hope this is now possible to review.
Abstract: would like to see the numbers for the reliability statements in the results	These are now included.
PMOS covers In-hospital care-> please describe somewhere in aim.	This is now included in the objectives.
Table 1: the gender numbers do not add up to 100%. Are all these others excluded? The other 2% missing?	This has been corrected.
The discussion gives the recruitment rate, however this is not mentioned in the results. Why not? Please add response/ recruitment rate to the results section of both questionnaires.	Thank you for highlighting this. For PMOS-30 as data came from a quality improvement project as opposed to primary research data collection, response rates were not collected. Therefore we are only able to present the recruitment rate for PMOS-10 validation element. The PMOS-44 recruitment rate (from the trial) is referred to in

	the introduction (86%). We have added the PMOS-10 recruitment response rate to the results section.
Typos: p18, line 25 -> two 'to's' to enhance P18 line 50-> two 'of's'	Thank you for highlighting this.
References: there is definitely more recent material than Vincent 2002; a lot has happened in patient safety since then.	Thank you – we have added two further references here.
I do not see a checklist (COSMIN would be a relevant one; although only relevant parts)	Thank you for highlighting this. We have included the relevant parts of the COSMIN checklist.
I cannot find any funding information	This is now included.
Reviewer 2	
P 5, Objectives, line 27-35. Formulation of this section so that it matches the headline (Objectives) would be welcome, as it stands now it can be perceived as methods. And in this form it seems more appropriate as meta text at the beginning of the Methods section, where an introduction to “what kind of study was carried out” is missing momentarily	Thank you for highlighting this, we have re-worked this section as suggested.
P 5 onwards, Methods section. A description of the PMOS-44 is missing, how many items/domains, how many negatively worded items? what kind of answer categories, how is the scores and result calculations?	We have included a description of the response options for all the PMOS measures in the method. The number of negatively worded items for PMOS-44 is referred to in the results section: ‘To facilitate the likelihood of reliable and valid responses, we also reduced the quantity of negatively phrased items in the revised measure. In the original measure the proportion of negatively phrased items was 57% (25 items); in the revised measure this was reduced to 33% (10 items)’ We could include the number of negatively worded items for PMOS-44 in the measure section, however we would also have to present the number of negatively worded items for PMOS-30 here too – which we feel sits better in the results section. Due to word count limitations it would be difficult to include a full description of PMOS-44 covering the items and linked domains. Therefore we link to papers relating to the development and assessment of PMOS-44. The domains are referred to in the introduction section.
P 6, Table 1 – is it possible to add the year of data collection? This would clarify	This is now included in Table 1.
P 6, Analysis onwards. I would welcome a clearer description of the criteria set prior to the analysis for evaluation psychometric properties of the PMOS-30 and PMOS-10. This would ease the reader and give a firmer and more trustworthy base for	Thank you for the opportunity to clarify this. We have included how the item analyses of interest (standard deviation and inter-item correlations) were agreed prior to the consensus group meetings. The results of these analyses were brought to the consensus meetings along with the reflections from the researchers/research nurses who facilitated data collection in the PRASE RCT.

the evaluation and conclusions. E.g. P 7 line 30: “We used multiple criteria, rather than only a statistical approach...” Please clarify both types of criteria somehow.	The reference to: “We used multiple criteria, rather than only a statistical approach...” relates to the activity in the consensus group meetings where we deliberately did not weight criteria (i.e. researcher/research nurse reflections or item analyses), and for each candidate item considered all criteria collectively. We felt this was appropriate as the PMOS-30 needed to retain the diagnostic function and link to domains and theory in the context of the wider PRASE intervention. For the PMOS-10 the shortening exercise followed a previously reported method by Marteau & Bekker.
P 14, Methods, Participants, Stage 1. I miss mentioned the number of participants, not only a reference to table 1	This is included at the bottom of page 5.
P 14, Methods, Participants, Stage 2. In this section you mention 165 patients recruited across nine wards. A description of the setting seems needed in general	Thank you for highlighting this. The spread across settings is now included in Table 1.
P 15, Measures. PMOS-10 This section seems to fall apart, is some text missing? It is not clear why the FFT is presented, it becomes clear later (P 20, line 16-18), but then actual technical information (number of items, domains, answer categories, scoring etc. is missing)	Thank you for the opportunity to revise this. We have made it clear why FFT is presented and included this earlier in the measures section. In the measures section we have also included the PMOS responses categories.
P 18, Discussion. Line 42, “Five domains.....” This sentence and the further needs clarification, e.g. Cronbach’s alpha was only calculated for 2 of the five domains, due to the item numbers in the domains.	Thank you for highlighting this, we have clarified this section of the discussion.
P 19, Strengths and limitations; “there was also a number of researcher and hospital volunteers with an extensive amount of...” if this is highlighted as a strength, it is needed to know the volume of “a number of researcher and hospital volunteers”. There seems to be only one limitation in this study, usually survey-based studies have a number of limitations. Given these have been overcome, it might be worth mentioning for others to learn from. The implications of the strength and limitations are not clear to the reader, please specify.	Thank you for highlighting this. We have added further clarification on the first point. We have also added further limitations, and attempted to clarify the implications.
Last comment, I do not find the supplementary reporting (e.g. trial registration; funding details; CONSORT, STROBE or PRISMA checklist)	Thank you for highlighting this. We have included the relevant parts of the COSMIN checklist.

VERSION 2 – REVIEW

REVIEWER	Marije van Melle Cambridge & Peterborough Foundation Trust University of Cambridge UK
REVIEW RETURNED	02-Sep-2019

GENERAL COMMENTS	Thank you for the opportunity to review this interesting article on an important topic. The manuscript has improved a lot. The methods are a lot clearer and more structured. Description of the validation of both questionnaires is more complete making it easier to read and better reproducible. Some minor details/ very minor revisions: - Patient and public involvement statement: I would summarise the involvement in this paragraph; e.g. Patients and public were involved in this study in the shortening and rewording of the questionnaires.....- Table 1 Characteristics: The age and length of stay in days are presented in mean and SD. However, these are rarely normally distributed, and I doubt they are here considering the broad SDs. I would suggest presenting median and range/ IQR here.- I still see a few footnotes; I would prefer this information to be in the text, or for footnote A to go in the references.- limitations: I think the amount of missings (mainly not applicable) is high in the longer questionnaires. Excluding them listwise (As I understand these patients are entirely excluded from all analyses) might have an effect on the generalisation. Maybe this could be addressed in the limitations. Another option (for the PMOS30) would have been to exclude this patient only in the dimensions that have missings and not the entire patient.
---

REVIEWER	Solvejg Kristensen Aalborg University Hospital - Psychiatry Mølleparkvej 10 9000 Aalborg Denmark
REVIEW RETURNED	08-Aug-2019

GENERAL COMMENTS	Thank you for the opportunity to re-review this paper. It has improved with your revisions, and I'm happy to recommend the paper accepted.
--

VERSION 2 – AUTHOR RESPONSE

Reviewer: 1

Thank you for the opportunity to review this interesting article on an important topic.

The manuscript has improved a lot. The methods are a lot clearer and more structured. Description of the validation of both questionnaires is more complete making it easier to read and better reproducible.

Thank you for this feedback, and for your feedback on earlier versions.

Some minor details/ very minor revisions:

- Patient and public involvement statement: I would summarise the involvement in this paragraph; e.g. Patients and public were involved in this study in the shortening and rewording of the questionnaires.....

We have made this change.

- Table 1 Characteristics: The age and length of stay in days are presented in mean and SD. However, these are rarely normally distributed, and I doubt they are here considering the broad SDs. I would suggest presenting median and range/ IQR here.

We now include median and range instead of mean and SD.

- I still see a few footnotes; I would prefer this information to be in the text, or for footnote A to go in the references.

We have removed footnote A as the two supporting references provide enough information about the wider improvement project.

- limitations: I think the amount of missings (mainly not applicable) is high in the longer questionnaires. Excluding them listwise (As I understand these patients are entirely excluded from all analyses) might have an effect on the generalisation. Maybe this could be addressed in the limitations. Another option (for the PMOS30) would have been to exclude this patient only in the dimensions that have missings and not the entire patient.

Thank you for highlighting this. We now reflect on this in the limitations section.

VERSION 3 – REVIEW

REVIEWER	Marije van Melle Cambridge & Peterborough NHS Foundation Trust & University of Cambridge
REVIEW RETURNED	29-Sep-2019
GENERAL COMMENTS	Thank you for the opportunity to re-review this article and for the accurate rebuttals. I am happy to recommend the article to be accepted.